# BMJ Open | Evaluating recovery following hip fracture: a qualitative interview study of what is important to patients

Frances Griffiths,[1] Victoria Mason,[1] Felicity Boardman,[1] Katherine Dennick,[2] Kirstie Haywood,[3] Juul Achten,[1] Nicholas Parsons,[1] Xavier Griffin,[1] Matthew Costa[1]

▶ Prepublication history and additional material is available. To view please visit the journal (http://dx.doi.org/10.1136/bmjopen-2014-005406).

[1]Warwick Medical School, University of Warwick, Coventry, UK
[2]Florence Nightingale School of Nursing and Midwifery, King's College London, London, UK
[3]Royal College of Nursing Research Institute, University of Warwick, UK

**Correspondence to**
Professor Frances Griffiths;
F.E.Griffiths@warwick.ac.uk

## ABSTRACT

**Objective:** To explore what patients consider important when evaluating their recovery from hip fracture and to consider how these priorities could be used in the evaluation of the quality of hip fracture services.

**Design:** Semistructured interviews exploring the experience of recovery from hip fracture at two time points—4 weeks and 4 months postoperative hip fixation. Two approaches to analysis: thematic analysis of data specifically related to recovery from hip fracture; summarising the participant's experience overall.

**Participants:** 31 participants were recruited, of whom 20 were women and 12 were cognitively impaired. Mean age was 81.5 years. Interviews were provided by 19 patients, 14 carers and 8 patient/carer dyad; 10 participants were interviewed twice.

**Setting:** Single major trauma centre in the West Midlands of the UK.

**Results:** Stable mobility (without falls or fear of falls) for valued activities was considered most important by participants who had some prefracture mobility and were able to articulate what they valued during recovery. Mobility was important for managing personal care, for day-to-day activities such as shopping and gardening, and for maintenance of mental well-being. Some participants used assistive mobility devices or adapted to their limitations. Others maintained their previous limited function through increased care provision. Many participants were unable to articulate what they valued as hip fracture was perceived as part of their decline with age. The fracture and problems from other health conditions were an inseparable part of one health experience.

**Conclusions:** Prefracture mobility, adaptations to reduced mobility before or after fracture, and whether or not patients perceive themselves to be declining with age influence what patients consider important during recovery from hip fracture. No single patient-reported outcome measure could evaluate quality of care for all patients following hip fracture. General health-related quality of life tools may provide useful information within clinical trials.

### Strengths and limitations of this study

- The study sample was representative of the age profile, gender balance and dementia levels of NHS patients experiencing hip fractures.
- It is possible that those not agreeing to be interviewed were struggling most with recovery.
- The data are limited by the difficulty the more physically and cognitively impaired patients had in giving a detailed account of their health experience.

global incidence of 1.31 million was reported and was associated with 740 000 deaths.[1] Hip fractures constitute a heavy socioeconomic burden worldwide. The cost of this clinical problem is estimated at 1.75 million disability adjusted life years lost, 1.4% of the total healthcare burden in established market economies.[1] Among those experiencing fragility hip fracture in England, Wales and Northern Ireland, 70% are aged 80 years or older, 73% are women and 34% are cognitively impaired preoperation. The mortality rate within 30 days of operation was 8.2% in 2013.[2]

The NHS has identified the need to evaluate the quality of service provision for patients with a hip fracture; this evaluation is conducted through the National Hip Fracture Audit Database (NHFD).[2] Currently, aspects of care such as time to surgery, length of patient stay and patient mortality in hospital and 30-day and 120-day follow-up are recorded in the NHFD. These data are now used to guide payments to healthcare providers, the payment being increased if the provider supplies 'best practice' care.[3] However, while important, there is interest from policymakers in the potential to enhance these currently reported data fields by including an assessment of outcome as reported by patients. It is increasingly expected that healthcare evaluations should include domains of health that are important to patients,[4] captured by well-developed patient-reported outcome measures

## INTRODUCTION

Fragility fracture of the proximal femur (hip fracture) is one of the greatest challenges facing the healthcare community. In 1990, a

(PROMs) which aim to assess how patients function and feel in relation to a health condition or associated treatment.[5] PROMs capture information that cannot be obtained by other means,[5 6] complementing more traditional performance or process-based measures.

Our aim was to establish whether or not one PROM could be used with all patients who experience a fragility hip fracture as part of the evaluation of the quality of healthcare for hip fracture delivered by the NHS. For this patient group, we were unable to identify a PROM specific to the assessment of hip fracture, and robust evidence of the quality and acceptability of non-hip fracture-specific PROMs following completion by patients sustaining a hip fracture is limited.[7] Moreover, clarity with regard to the outcomes of healthcare that these patients consider relevant and important does not exist. Appropriate and relevant PROM-based assessment should be underpinned by an understanding of what is important to patients in terms of the outcomes of healthcare. Further, we were concerned to understand whether, for people with different prefracture health and social context, what was important to them during recovery was different. For example, we hypothesised that what is important to a younger, otherwise healthy person experiencing hip fracture may be different from what is important to a person who perceives themselves as nearing the end of life. Good quality care would, as far as possible, enable each patient to achieve what is important to them in terms of recovery. If a PROM is to be used to assess quality of care, the measure needs to capture this. We therefore designed an interview study to explore with patients and, where appropriate, their carers, what they consider to be important outcomes and to explore variation across this patient group. Our research questions were:

1. What do patients who have recently experienced a hip fracture consider important when evaluating their recovery?
2. Is there variation between people within this population of the experience of what is considered important in recovery from hip fracture and why?

These research questions are framed by the desire of policymakers to evaluate the quality of care for hip fracture through assessment of recovery from the perspective of the patient.

## METHODS
### Study design
We conducted semistructured interviews with patients and, where appropriate, their carers at two time points, at approximately 4 weeks and then again at 4 months after they had sustained a fragility hip fracture.

### Identification of patients with a hip fracture
We recruited participants from an existing cohort study, the Warwick Hip Trauma Evaluation,[8] that started in January 2012. This is a cohort of all patients admitted with a hip fracture to a single major trauma centre in the West Midlands of the UK. As part of their preoperative assessment, patients were assessed for their capacity to consent using clinical assessment and the Abbreviated Mental Test Score (AMTS).[9] The AMTS is a 10-item measure used to rapidly assess the possibility of cognitive impairment in elderly people. A score below 8 suggests cognitive impairment.[10] Scores less than 8 were taken to indicate that a patient was unlikely to be able to consent for themselves. Those deemed to have capacity for consenting to surgery, based on clinical assessment and AMTS, were considered able to consent for this study. Following the emergency surgery for their fracture, those with capacity gave written consent to be approached for interview. For those deemed not to have capacity due to cognitive impairment, verbal consent was obtained from their consultee.[11]

### Sampling
During the data collection period for this study, February to August 2012, we purposefully sampled cohort participants who had reached 4 weeks or 4 months following their hip fracture and had consented to be approached for interview. The time points were chosen to be the same as those used for data collection for the NHFD.[12] If a PROM were to be used with this patient population to assess quality of care, patients would be asked to complete the PROM at these time points. Our sampling strategy ensured a diverse mix of patients with respect to the following factors: age, gender, AMTS[9] and EQ-5D score.[13]

### Interview recruitment and consent process
We contacted eligible patients and carers by telephone just prior to 4 weeks and/or 4 months following hip fracture first to invite them to be interviewed, then to arrange an interview. If patients declined to participate, the reasons offered were recorded. Patients with capacity to consent were contacted directly. For those patients deemed not to have capacity, we contacted their consultee. Patients able to consent for themselves signed their own consent forms. For those unable to consent, the consultee signed an agreement form and we aimed to interview a carer as well as the patient (patient/carer dyad). Carers who were interviewed signed a consent form. Initial analysis started during the recruitment phase; recruitment continued until data saturation at the first time point. The study flow diagram is at figure 1.

### Interview process
We interviewed participants at their current residence (own home, residential or nursing home) or in hospital. The interviewer was trained in interviewing but did not have clinical knowledge of hip fracture, its treatment or prognosis. Where possible, patients and carers were interviewed alone; however, where the carer and patient requested a joint interview (whether or not the patient had cognitive impairment), they were interviewed together. The aim of the interviews was to understand each participant's lived experience of hip fracture[14] and

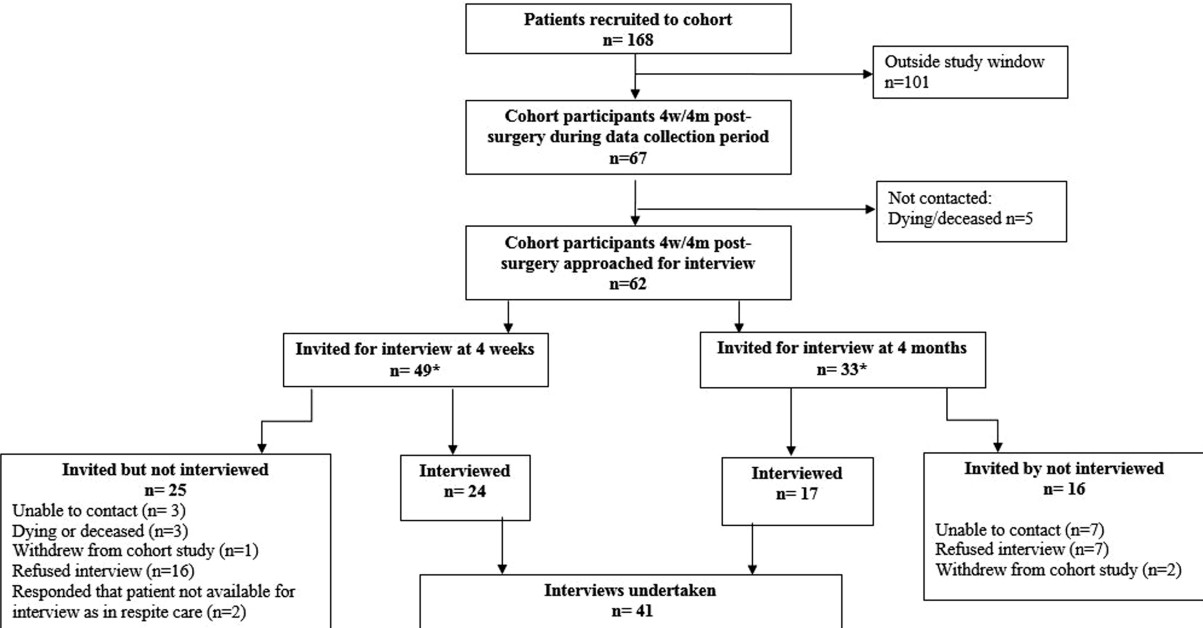

**Figure 1** Flow chart of study recruitment.

the influence of their social context and prefracture health. We used the following questions:

▸ What is a normal day like for you now?
▸ How bothersome are you finding your hip?
▸ What is different about your life now compared to just before your injury?
▸ Compared to just before your injury, what has stayed the same?
▸ Which of these make the most difference to your life?

The interviewer encouraged participants to talk about the experience in whatever order they chose and using terms meaningful to them. Later in the interview we prompted, where necessary, for clarification about what in the patient experience was related to the hip fracture. Towards the end of the interview, we directly asked what was important to them in terms of recovery if this had not already been talked about by the participant, using the following questions:

▸ What is important to you in terms of your recovery?
▸ Where would you like to see yourself in the future in relation to your recovery (ie, the next few weeks and months)?
▸ If a friend or neighbour were asking you now about how well you are recovering, what has been important to you that you would tell them about?
▸ If a doctor or nurse was asking you now about how well you are recovering, what would be important for the doctor or nurse to ask about?

Consideration was given to the potential challenges associated with interviewing older adults, for example, by giving potential participants sufficient time to decide whether or not to participate and minimising burden and fatigue through streamlining questions.[15] The interview process, questions and prompts were refined by the study team during the initial stage of data collection,

particularly adding questions and prompts to focus the participant on recovery from their hip fracture. Questions were similar for both the patient and the carer. Interviews were audio-recorded. For one interview, an audio recording was not feasible due to the noisy environment, so extensive field notes were taken. For all interviews, the researcher made reflective field notes to assist interpretation of the interview data.

## Analysis

Interviews and field notes were transcribed and transcripts checked, anonymised and uploaded into the Nvivo software.[16] Initial analysis involved data immersion, reading and re-reading each transcript and discussion of the interview transcripts by the research team. Our research team was multidisciplinary: social science, behavioural science, health science, orthopaedic surgery and statistics. All team members read at least five transcripts, so all transcripts were read by at least two team members. From the data, we identified and crystallised what was important for participants that was specific to hip fracture recovery.[17] We found that the interviews at 4 weeks and 4 months covered very similar issues, although, as would be expected, what the participants reported about each issue 4 weeks and at 4 months was different, as recovery was more advanced at 4 months. As our analysis aimed to identify what patients consider important when evaluating their recovery rather than the detail of recovery itself, we treated all the interviews related to one participant as one set of data. During data interpretation, we took account of the timing of the interview, whether the interview data were from a patient or carer or patient/carer dyad, and field notes.[17] For data collection and analysis, we took a phenomenological approach in that we sought to understand the participant's experience of hip fracture recovery and the influence of

their context on this,[14] [18] and concurrently we took a selective realist position[19] in that we recognised hip fracture as an event identifiable by means other than through the participant's account.

We used two different approaches to analysis to answer our research questions. For the first research question, which is concerned with the whole groups of participants, we used thematic analysis.[20] We searched the transcripts for any mention by the participants of what was important to them during recovery from hip fracture. These were discussed at team analysis meetings. Transcripts were then coded in NVivo. As coding proceeded, we reviewed these codes at our team analysis meetings and combined them into themes. After we had read, discussed and then coded 10 transcripts, we found no additional themes in the remaining data. Double coding was undertaken for one in four transcripts and coding compared and discussed to check consistency of final coding. During analysis, we became aware that although the data from different participants could be coded under the same theme as mobility, the experience of recovery was very different for different people. This led us to our second research question and analysis approach.

To answer our second research question, we used cross case analysis.[21] We considered each participant as an individual 'case' living within their particular context[22] [23] and through comparison of cases sought to understand how they varied. To develop our matrix for the cross case analysis,[21] we closely read five participant data sets and then developed, from the data, a template for summarising the experience of hip fracture recovery for each patient carer dyad. This involved considering each set of interviews as a whole, reading and re-reading the text and writing a summary of the patient/carer journey and all that influenced it. We reviewed the summaries at our data analysis meetings and from these initial summaries we developed a draft template. We refined the template based on the data as we summarised and discussed further transcripts. The template included: current and recent past living arrangements and environment, day-to-day life now and in the recent past, the impact of the hip fracture and its management, what was changing in day-to-day life as they recovered, the extent to which the patient referred specifically to the fracture and their ability to engage in the interview. Each of these formed a data row in our matrix with a column for each participant. The data about each patient were summarised into the template with a second research team member reviewing each summary against the data. To qualitatively understand the variation in the experience of what was considered important for recovery, we compared these summaries.

## RESULTS

Twenty-one patients were interviewed on one occasion and 10 were interviewed twice giving a total of 31 patient participants and 41 interviews. Of the 31 patient participants, 20 (64.5%) were women, the mean age was 81.5 years (SD 9.2, range 61–96) and 12 (39%) scored less than 8 on the AMTS. Of the 41 interviews, 24 were conducted 3–9 weeks after the hip fracture, and 17 were conducted 14–23 weeks after the hip fracture. Nineteen interviews were with the patient only, 14 with the carer only and 8 with the patient/carer dyads. Interviews lasted between 20 and 90 min. Despite framing the interview for interviewees as exploring the experience of hip fracture, many interviewees talked about general health issues. Although we prompted to clarify what was related to their fracture, in many interviews it was difficult to disentangle the impact of the fracture from the impact of other health problems. Some interviews contained almost no data that were clearly related to the fracture. From the perspective of the patient, all their health problems were part of one experience. The absence of data clearly related to the fracture was more marked in the 4-month interviews compared with the 4-week interviews. We therefore decided not to attempt interviews at 12 months postfracture as originally planned.[8] The following sections report our analysis. Illustrative quotations from data are labelled with the age and gender of the patient, time since hip fracture and whether the quotation was from the patient or carer.

### What is important to patients when evaluating their recovery?

From our systematic search of the interviews for data related to recovery from the hip fracture, we identified the following themes: mobility, valued day-to-day activities, self-care, pain, mental well-being, fear of falling and leg shortening. When talking about mobility, day-to-day activities or self-care, participants also talked about their level of independence.

#### Mobility

This was the most prominent theme, although when talking about mobility the interviewees often mentioned other themes. Mobile participants reported limited mobility in the weeks postoperation and valued any improvement.

> I'm walking with a walking stick at the moment. I've been down the park and back…I can usually get around [the house] without the walking stick, and I can get up and down stairs no problem. I get upstairs with my good leg and downstairs with my bad leg. (Participant 6, male, age 78, 5 weeks post operation)

By 4 months, for many participants mobility had improved, and they were happy that they were returning to normal mobility.

> I can't rush round like I did, but eventually that will come…I mean it's pretty normal now, but I think it's going to be a while before I can actually walk as I did and I probably won't walk as I did…when I came home [from hospital] I was still hobbling…but now I'm more

or less…walking normal, especially with the stick. (Participant 10, female, age 83, 18 weeks post operation)

For those with limited mobility before hip fracture, any unaided improvement was limited to the prefracture level but also valued.

> The operation was successful and got him back to normal right from the start, right from the very first day that he had it done. He was able to then walk pain free with a Zimmer frame to the toilet. The staff were all saying it was amazing how well he was walking and he would soon be back to normal, but what they didn't realise was that he was walking normally. (Carer of participant 1, male, age 84, 16 weeks post operation)

Other participants were using mobility aids that they had not been using regularly before the fracture. For some, the addition of mobility aids enabled greater security of mobility than prior to their fracture.

> Her mobility's getting better. I think she'll cope with the frame. She's had a couple of falls in the home, earlier when she was forgetting that she had to use the frame. She'd get out of bed and not use the frame and consequently fall. But she's got in the habit of using it now… she's not falling, which is a bonus. (Carer of participant 13, female, age 87, 14 weeks post operation)

### Valued day-to-day activities

Those who were active prior to their fracture talked about the frustration of the restriction in their activities particularly in the weeks following the fracture.

> I'm back on what I call domestic duties—washing up! But the thing that is frustrating is that I can't get outside and do any gardening. (Participant 12, male, age 78, 6 weeks post operation)

> I just miss getting up and getting out. I never stayed in. I'd go out in the morning and come back and then I'd go out again, I just used to go out looking round the shops. I just get these crossword books and I do those. (Participant 20, female, age 92, 5 weeks post operation)

Participants who were active before their fracture were usually able to resume valued activities but had some limitations which remained a frustration.

> I can do little (gardening) jobs but because I haven't got as much movement in the hip joints, I find it difficult to go down on my hands and knees…If I go down on one knee it's difficult to get up again so that's not possible but I can do things that are higher up, I can trim. (Participant 15, female, age 61, 15 weeks post operation)

> I'm tackling a little bit of cooking now. I started to cook myself some nice lunches and I haven't got round to the…scones…I made one lot when I came home and I thought, I can't be bothered anymore. (Participant 10, female, age 83, 18 weeks post operation)

Some participants returned to valued activities through adapting how they did them, this participant using a wheelchair for the first time.

> Over the last three weeks, when we go out shopping now, I can't go down the aisles, so [daughter] gets me a (wheel)chair and I can sit in the chair and then say what shopping I need, that is very good. (Participant 9, female, age 92, 18 weeks post operation)

Participants who no longer undertook valued activities that involved significant mobility were content to continue as they were, for example, occupying themselves with visits from family and reading.

### Personal care

Washing, dressing and getting to the toilet was talked about in interviews, but in many cases it was not clear whether difficulties with personal care were specifically due to the fracture. A few interviewees talked about problems with incontinence, but again it was unclear whether this was specific to the fracture. Most patients had a commode or had arranged to sleep near the bathroom in the weeks immediately after the fracture. Some participants were able to describe problems with self-care specific to the hip fracture.

> I'm…not able to put a sock or anything on my injured leg. I can manage now with my trouser leg and throw these jogging trousers and hook my leg into them but I have to ask my husband if I need to put a sock or a shoe, or my slipper on that foot. (Participant 15, female, age 61, 6 weeks post operation)

At the second interview, this participant was pleased to report that she now needed very little help with self-care, at least in part through wearing alternative footwear.

> I still have to throw my clothes and hook them onto the foot to get dressed. I couldn't wear lace-up shoes or anything like that because I couldn't tie them up, but things like slip-ons and sandals I can get on quite easily, so I'm fairly independent—I am independent really, I just need help with cutting my toenails and that—those on the right foot that's all. (Participant 15, female, age 61, 15 weeks post operation)

### Pain

Although pain was talked about by some interviewees, it was not considered a major problem.

> So here I am, four or five weeks [post operation], I get a little bit of pain, not a lot. (Participant 7, female, age 70, 5 weeks post operation)

> The pain was so bad before I had it done, and I just couldn't believe the relief after the operation when I was walking in the hospital and I had one of those pushers you know. And there was no pain. And I kept thinking, I can't believe this, and that's how it's been. I've never

had any pain, not at all. (Participant 10, female, age 83, 18 weeks post operation)

There's several times, like when I have got to get up those steps. I put my right foot first and bring my left foot up, and once or twice…you step on your left, and it's still there, lets you know it's still tender. (Participant 12, male, age 78, 16 weeks post operation)

### Mental well-being

Low mood or depression associated with the reduced mobility due to the fracture was reported by a few interviewees, emphasising the great value placed by interviewees on being independently mobile.

He can't walk and that, to him he'd rather die. I'll be honest with you he's said it once or twice, "Let me go". And I said, "No you're not going no-where". And then the other day for the first time, but he hasn't said it since, "I'm going to commit suicide", I said, "No you're not, you're not". (Carer of participant 31, male, age 84, 5 week post operation)

For me it was a massive problem and caused me depression. To me is the most important thing, the mental aspect of taking away somebody's freedom to be able to move around and go to the shops and do all that sort of thing. (Participant 7, female, age 70, 23 weeks post operation)

### Fear of falling

The experience of the fracture left a few participants with a fear of falling and sustaining a further fracture.

I think it frightened him more than anything else. He's frightened he'll fall over again and do it again, that bothers him more than anything else. Because now when he stands up at all to try and walk he's frightened he's going to fall over and the same thing will happen all over again. (Carer of participant 11, male, age 84, 7 weeks post operation)

I've got to watch what I'm doing. If I catch my foot on [paving stone], I can go over again. (Participant 12, male, age 78, 16 weeks post operation)

The fear of falling was sometimes expressed by a family member. When talking about his frustration at not being able to work in the garden, participant 6 added

All the rain has made it very slippery, and [wife] says, "No way do you go out there." (Participant 12, male, age 78, 6 weeks post operation)

This emphasises the value given to mobility without falls or fear of falls by interviewees.

### Leg shortening

This is a problem that is common following extracapsular fracture of the proximal femur. One interviewee described her concerns about this.

One leg is now shorter than the other so that makes walking a bit difficult because it gives me back pain. (Participant 15, female, age 61, 15 weeks post operation)

### Is there variation within this population of the experience of what is considered important in recovery from hip fracture?

Our sample included patients from across a spectrum that extended from those who were physically and mentally active prior to their fracture through to those who, prefracture, had been immobile due to conditions such as multiple sclerosis, chronic obstructive airways disease and arthritis, and those with severe cognitive impairment. Although when talking about what was important to them when evaluating their recovery from hip fracture, patients from across this spectrum talked about similar themes, their experiences of what was important was different for different people. In box 1, we present condensed versions of the interview summaries developed during our second analysis approach, for participants chosen to represent the whole spectrum of patients. We indicate whether the data were provided by the patient, the carer or both.

### Recovery as a return to the prefracture state or as part of ageing and decline

Every patient interviewed had experienced a hip fracture and surgery, so in physical terms all of them had, for a period of time, been somewhat impaired compared with their prefracture state. Four weeks postoperation, those who were active prefracture talked in terms of regaining a recovered state that was similar to their prefracture state, though with some minor adaptations (participants 15 and 20 in box 1). While these participants expressed worry about how well they might function in the future, there was, nevertheless, determination to progress to as full a recovery as possible. Four months postoperation, many of these participants had all but regained their prefracture level of activity. Among participants with severely limited mobility prefracture, some were able to identify specific activities which were more difficult postfracture than prefracture, such as putting on socks and getting in and out of bed. Some were also able to identify specific improvements in mobility postoperation (see participants 9 and 15 in box 1). These participants described a process of recovery, although it was very limited.

In contrast, for other participants, the fracture was just one part of a process of ageing and decline. For example, participant 11 (see box 1) had been very limited in his activities before the fracture. Postfracture, he needed adaptations to his home and increased care support postfracture to enable him to continue to manage at home. The mobility of participant 18 had declined and she had started using a wheelchair instead of her mobility scooter to get out of the house. However, it was unclear whether the decline was due to the concurrent heart failure or the fracture. Those who were the most physically or cognitively impaired prefracture did not talk about regaining a recovered state but about a state of no change. They

---

**Box 1**  Summaries of the data about individual patients and their recovery from a hip fracture

A 61-year-old female social worker, who lives with her husband. Before her fracture, she was working full time and, for recreation, taking country walks, undertaking all types of gardening activities and playing with her grandchildren. Postfracture fixation (6 weeks), she described using crutches to get around the garden and shops, needing help with putting on socks and cutting toe nails, and was unable to climb stairs. She talked in terms of improvement and expectation of returning to work and full activity including cleaning and gardening. By the second interview, she was frustrated that recovery was so slow, but she could identify the ways in which she had continued to recover. (Participant 15, interviewed 6 and 15 weeks postoperation).

A 92-year-old woman, who lives alone in her own flat within a sheltered housing complex. Prior to the hip fracture, she looked after herself and did her own washing, but had a cleaner to undertake heavy household chores. She spent most of each day out and about at the shops, engaging in social activities, bingo and on outings. She had no other illnesses. Postfracture fixation, she talked about having some initial pain and problems lifting her leg after the operation but was now mobile about her home with a walking frame. The housing complex has a lift which she now used. She was intending to return to getting out and about as she was before her fracture. (Participant 20, interviewed 5 weeks postoperation).

A 92-year-old woman, who lives with her husband. Her daughter visits several times a week to help. Poor hearing. Difficult to disentangle what was before and after fracture. Seems to have been able to walk around the house, undertake self-care and microwave own meals pre-fracture. Postfixation of the hip fracture, patient slowly improved walking. Life seems very similar to before fracture except for the need for a walking aid, inability to put on socks and husband now microwaves the meals. (Participant 9, interviewed 9 weeks postoperation).

A 70-year-old male retired painter and decorator, who lives with his wife and enjoys almost daily visits from his grandchildren. Mobility restricted to 5–6 m for more than 2 years prior to fracture due to knee pain and chronic obstructive pulmonary disease. When interviewed, he describes struggling to get up the stairs, get in and out of bed, put his shoes and socks on, and bend down. Although his mobility was severely restricted prior to his fracture, he described being unable to get around as much as he had done before the fracture. He noted some improvement over recent weeks, as he no longer needed two sticks for walking, only one. (Participant 3, interviewed 15 weeks postoperation).

An 84-year-old man with dementia, who has some lucid moments and some recall of falling and hurting himself. He lives with his wife who looks after him and they have a cleaner to do heavy housework. His wife provided the interview, involving the patient in the latter half when he woke up. The patient's walking was gradually slowing and he had a number of falls before his fracture. The fracture occurred while walking in the shopping area with his wife. Since fixation of the fracture, the patient has required assistance with personal care, has professional carers four times a day, and the bathroom has been adapted for his limited mobility. The interviewee had difficulty distinguishing decline due to old age and change due to the fracture. The patient presented with some pain but it was unclear whether this was from the fracture or previously established osteoarthritis. Before the fracture, both the patient and his wife had ceased all non-essential activities except for a weekly trip to the shops, so daily life had changed little except for more care provision. (Participant 11, interviewed 7 weeks postoperation).

A 74-year-old woman, who lives with her husband. The patient lived with severe rheumatoid arthritis for 30 years. Developed heart failure and was admitted to hospital with shortness of breath and confusion. Fell while in hospital and fractured her hip. Mobility before hip fracture very limited—able to walk slowly in house and garden, undertake light chores and use the scooter to go shopping. Became worse with breathing difficulty. Mobility remained reduced after hospital admission. Able to take steps slowly in house with support. Uses wheelchair to go out of house—a new ramp improved this by the second interview. Unclear how much mobility change was due to the fracture and how much was due to heart failure. (Participant 18, interviewed 6 and 18 weeks postoperation).

An 88-year-old female retired teacher, who lives with her son and has a diagnosis of multiple sclerosis. The patient wove together preinjury and postinjury experience in her account, making it difficult to disentangle. She said her son does the cooking and cleaning and her daughter assists with self-care. She has a close family, feels well supported and has lots of visitors—friends, grandchildren and great grandchildren. Her main interest beyond seeing friends and family is reading. She described being content with life. Prior to her fracture, she was unwell with an infection and recounts using a frame for mobility which she still uses. (Participant 23, interviewed 5 weeks postoperation).

An 85-year-old woman, who lives in a nursing home. Her daughter visits on alternate days. Her daughter provided the interview data. The patient has dementia but otherwise had been well before the fracture. She gets up and walks about herself, and takes herself to the toilet. She enjoys sitting and chatting. The patient does not remember the injury. Her life has not changed from how it was preinjury. The daughter did not mention any fracture-specific issues related to recovery. (Participant 26, interviewed 6 weeks postoperation).

An 84-year-old woman with limited English language. Preinjury, she had carers to assist her with all her personal needs. The injury had occurred while being hoisted. Postinjury, her main concern was that at discharge from hospital, after a 3-month stay, she was sent to a nursing home where she knew no-one. The patient repeatedly expressed distress about being in the nursing home but did not talk about the fracture. (Participant 5, interviewed 18 weeks postoperation).

An 84-year-old man, who has dementia. He lives alone but received visits three times a day from his son who provides meals. The son was interviewed. Arthritis of the knee limited mobility before the fracture. Spent most of the day sitting. At weekends prior to the fracture, the patient went to his neighbour's house for an evening meal. The patient fell and sustained a fracture while walking to his neighbour's house. He does not recall the fracture. At the time of the interview, the patient was as mobile as preoperation, limited by pain and stiffness from arthritis. Not yet visiting neighbour, but this was because the family was discouraging this in case he falls again rather than due to mobility. (Participant 1, interviewed 16 weeks postoperation).

---

continued with their limited activities as before (for example: participants 23 and 26 in box 1). For one participant, the only change was her move to a new nursing home (participant 5 in box 1). Participants with cognitive impairment were often unaware of having experienced a fracture (participant 1 in box 1).

## Recovery through adaptation

In the face of their physical limitations, most participants made adaptations that mitigated the effect of the fracture; for example, employing a cleaner, moving to a nursing home or using a walking aid or other assistive device. For those who were active prefracture, adaptation was mostly considered temporary, although at 4 months there was some evidence that active patients had adapted to some limitations such as being unable to kneel for gardening or limiting time spent shopping to avoid exhaustion. For some participants who had been experiencing decline in their mobility prefracture, the fracture precipitated adaptations that they had not previously considered but made their life easier. These included using a wheelchair for shopping, having a new ramp built for getting in and out of the house in a wheelchair, using a walking aid or employing professional carers to assist with personal care. For some, their own or their carer's fear of further falls limited their mobility or at least limited how far they tested their ability to walk. Poor weather conditions exacerbated this fear, but adaptations to the environment such as walking aids or handrails lessened the fear.

## DISCUSSION

Following hip fracture, for those who had some prefracture mobility and were able to articulate what they value during recovery, stable mobility, that is, mobility without the experience of or fear of falling, and mobility that allows people to undertake valued activities are most valued. The ability to walk is important, but so too are other leg movements needed for activities such as gardening or using transport. For some participants, maintaining mobility, however limited, was achieved by using assistive devices or working out new ways of doing an activity. Some participants adapted to their limitations, for example, wearing different footwear or adjusting their expectations of what they could achieve. Others maintained their previous limited function through increased care provision.

Patients also consistently valued certain basic domains of health, such as pain (or lack of it), day-to-day activities, personal care and mental well-being. However, many participants in this study were unable to articulate what was important to them in terms of recovery from hip fracture. The hip fracture was just one part of their decline with age and its impact could not be disentangled from the impact of other health issues. The level of recovery perceived by a participant was influenced by their prefracture state and their ability to make adaptions during recovery.

## Strengths and weaknesses of the study

When the mortality rate postoperation is taken into account, including the higher mortality among older women, the study sample was broadly representative of the age profile and gender balance of the population of England, Wales and Northern Ireland experiencing hip fractures.[2] We used a higher cut-off for assessment of cognitive impairment (score of 8 on AMTS) compared with the NHFD (score of 6 on AMTS). This is likely to explain our higher proportion of participants with cognitive impairment compared with the average in the NHFD.

More research time was spent on recruitment than any other aspect of the study as it proved difficult. When contacted about the interview study, potential participants talked about other priorities or concerns that prevented them agreeing to an interview, or they simply did not wish to be interviewed. It is possible that those not interviewed were struggling most with recovery. Our data are also limited by the difficulty some frail older adults have in giving a detailed account of their health experience.[24] Interview data are jointly constructed by the interviewer and interviewee,[25] and our interviewer had no clinical knowledge of hip fractures. This reduced the likelihood of the interviewer influencing the data. A clinician undertaking the interviews would have the knowledge to help the patient tease out whether the health problems were fracture related or not. However, this would have obscured the important finding that participants often experienced their fracture as part of, rather than separate to, their other existing health problems. We relied on carer's accounts for some participants. We found that they talked about the same themes as the participants. However, for those with cognitive impairment, some carers were unable to provide detailed data as they had limited day-to-day contact with the participant. We did not attempt to check with participants about our interpretation of the data to avoid a further burden for them.

### Comparison with other studies

There are similarities between our findings and other qualitative studies of similar populations. A Swedish team that explored engagement with rehabilitation post hip fracture found a similar spectrum of participants.[26] They classified their participants as: those who were frail and in need of support but did not request it; those who were dependent and took no active part in rehabilitation and those who were self-sufficient. Another Swedish study, undertaken with people 12 months after their hip fracture, found that mobility and a return to normal activities were key outcomes for patients.[27] An Australian study of mobility postfracture found that reduced level of mobility was associated with a fear of falling, physical limitations from other illness and social/environmental factors.[28] Our results also echo findings from across the research literature on the experience of health and illness. For example, the difficulty disentangling the impact of one health condition from other comorbidities has been found for mental health conditions.[29] The acceptance of an acute health problem as being part of the ageing process has been found for conditions such as stroke.[30] Recalibration to altered circumstances in response to a sudden injury has also been described,[31] as have the

adaptations—both physical and psychological—that people make in order to maintain their quality of life.[32] Reduced expectations of health and acceptance of limited function have been described among elderly women.[33] Fear of falling is common among older people generally.[34] The consistency between our findings and other studies suggests that there is now sufficient qualitative evidence to inform policy decisions about the choice of appropriate PROMs for assessing recovery from hip fracture.

### Implications for clinicians and policymakers

This study was undertaken in response to a potential policy change involving the use of a PROM to assess patient recovery from hip fracture, the results of which would form part of the evaluation of the quality of care provided for hip fracture. We conclude that for the population experiencing fragility hip fractures, it is unlikely that a single PROM specific to hip fracture could be developed which is relevant to the whole spectrum of patients. An assessment that focuses on mobility of the hip would be relevant for many patients, and mobility impacts on other health domains. However, with any form of assessment of mobility, prefracture status would have to be taken into account. Some patients had limited prefracture mobility at the hip, so a lack of mobility during recovery may not reflect the quality of care. In addition, there are other factors that influence the perception of recovery by patients. These include adaptations that they or their carers make to compensate for their reduced mobility, and patient perception of whether or not they are at the stage in life where decline is inevitable. Quality of care is only one of a number of inter-related factors that influence the patient's perception of recovery from a hip fracture.

Several of the themes described by interviewees—mobility, day-to-day activities, self-care, pain and mental well-being—are similar to the domains included in currently available generic measures including the EuroQoL EQ-5D,[13] the Short Form 36-item Health Survey (SF-36)[35] and the WHOQoL-BREF.[36] Both the EQ-5D (3L) and the SF-36 (V.1) have been widely used in trials of people sustaining hip fractures, but for both measures evidence of essential measurement and practical properties is limited.[7] In the context of a clinical trial where patients are randomised to an intervention and control arm, these generic measures may be appropriate but they may need to be supplemented by specific tools for selected groups, such as patients with high levels of preinjury function.

**Acknowledgements** The authors thank Katie McGuinness, Kate Dennison, Zoe Buckingham, Catherine Richmond, Rebecca McKeown, Hayley Rice, Mamta Rana, Filo Eales and Gail McCloskey for their assistance in recruitment and all the patients for their time and effort in participating in this study.

**Contributors** MC, FG, JA, XG, KH and FB contributed to the conception and design of the study. FB, VM and KD conducted the interviews. All authors contributed to the analysis and interpretation of data. FG, KD and VM drafted the article and all authors revised it critically for important intellectual content. All authors gave final approval of the version to be published.

**Funding** This study was funded by a Programme Development Grant from the National Institute of Health Research (RP-DG-1210-10022). This study was co-sponsored by the University of Warwick and University Hospitals Coventry and Warwickshire NHS Trust. This manuscript presents independent research. The views expressed are those of the authors.

**Competing interests** None.

**Patient consent** Obtained.

**Ethical approval** Ethical approval was granted by NHS REC London—Camberwell and St Giles (11/LO/0927) on 18 August 2011. Further approval was obtained from the research and development department of the University Hospitals Coventry and Warwickshire NHS Trust. This research complies with the Helsinki Declaration.

**Provenance and peer review** Not commissioned; externally peer reviewed.

**Data sharing statement** No additional data are available.

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
