## [Reviewer comments · BMJ Open]

ARTICLE DETAILS

TITLE (PROVISIONAL)	Evaluating recovery following hip fracture: a qualitative interview study of what is important to patients
AUTHORS	Griffiths, Frances; Mason, Victoria; Boardman, Felicity; Dennick, Katherine; Haywood, Kirstie; Achten, Juul; Parsons, Nicholas; Griffin, Xavier; Costa, Matthew

VERSION 1 - REVIEW

REVIEWER	Yuchi Young State University of New York at Albany, New York, USA
REVIEW RETURNED	19-Jun-2014

GENERAL COMMENTS	The manuscript requires major revisions. It is not acceptable for publication in its present form. This study addresses an important issue related to wellbeing and evaluation of services to hip fracture patients. The objectives of this study are to (1) explore what is important to patients in postacute functional recovery among hip fracture patients who received surgical repairs, and (2) consider how these reported priorities could be used in advancing the evaluation of the quality of hip fracture services. The manuscript is confirmatory and may add new information to the hip fracture functional recovery literature from the perspectives of the patients, carers, and patients with dementia. Major revisions, however, may be needed to address the research questions. Minor issues 1. Literature updates: most of literature cited in this study were in or prior to 2012; only one article was in 2013. More current literature should be included.2. Figure 1 clarification: the flowchart does not provide a good summary of the Participant section. Figure 1 starts with the number of recruitments of study sample and ends with number of interviews. The following clarifications may be helpful for readers to follow the sample selection:a) The final number of study subjects was 31. Of these, how many interviews were conducted with individual patients, with individual carers, and the number of joint interviews conducted with carers responding for patients with dementia or too frail to be interviewed.b) Separation of those who were interviewed at 4 weeks or 4 months only, and those who were interviewed twice at 4 weeks and 4 months (n=10). Major issues
---

The study combined the data collected at 4 weeks and 4 months because the self-reported results are similar (page 8, Analysis, lines 15–20). The results from this combination of time related to functional recovery at approximately 4 weeks and 4 months may lead to biased results and may be less informative to address the second objective of the study. The data analysis plan may need to be changed to address the following issues:

1. Separate analysis of functional recovery at 4 weeks vs. 4 months may be needed.

Literature have shown that hip fractures have significant impacts on functioning related to the ability of self-care (ADLs) and independent living in the community (IADLS). These adverse outcomes mainly relate to compromised mobility function as a result of hip fractures. The process of regaining prefracture function can be lengthy and stressful. Most hip fracture patients were still very disabled at 4 weeks following hip fracture surgery and intensive postacute rehabilitation. Substantial peak function recovery was observed at approximately 4 months following postacute rehabilitation (a lot of literature available in Medline/Pubmed search), thereafter, the functional recovery slows down and level off. Lumping together data collected at 4 weeks and 4 months is less informative to study objective 2; that is, the priority considered important to the patients reported by the study can be used in the evaluation of the quality of hip fracture services (page 3, lines 10-11).

2. Separate data analysis by patients and carers.

Studies on validity and reliability of self-reported data have shown that bias may result from self-reported data. For example, hip fracture patients often underestimate their functional disabilities; on the other hand, carers—often very protective—overestimate the level of disability of the patient.

3. A homogeneity study may be performed to enhance generalizability of the study. The authors mentioned in the study Limitations section that patients who were unable to participate in the study may be older, sicker, and more disabled. Homogeneity tests on age, gender, selected health, or functional status may be performed to address whether the participants and non-participants were indeed different.

4. Results and interpretation should focus on addressing the two study objectives:

a) to clearly present what you found from the patients, carers, or patients/carers

related to the time frame in which data are collected – what results are confirmatory and what are contributions of this study to the literature.

b) the second objective of the study was less articulated. In page 16 lines 22-24 “we now have sufficient qualitative evidence to inform policy decisions about the choice of appropriate PROMS for assessing recovery form hip fracture.” The authors may want to quote specific evidence to support the statement. Other statements in Lines 33-35 and lines 54-56 in page 16 are appropriate for the study results: “it is unlikely that a single PROM specific to hip fracture could be developed which is relevant to the whole spectrum of patients,” “...it may be impossible to devise a single PROM that will be appropriate for all patients,” respectively.

Many of PROMS measures were originally designed to assess intervention or treatment effectiveness in clinical trials, but are now used more extensively to assess patient perspectives of care outcomes. This study results provide useful information to suggest that once a seemingly appropriate measurement has been identified,

	it is advisable to pilot the questionnaire on a small number of patients given the heterogeneity of hip fracture patients and the diverse priority list on recovery concerns generated from this study.
--	---

REVIEWER	Nicholas Taylor La Trobe University, Australia
REVIEW RETURNED	20-Jun-2014

GENERAL COMMENTS	Evaluating recovery following hip fracture: a qualitative study of what is important to patients This qualitative study explored what patients and their carers consider important when evaluating their recovery from hip fracture. A total of 41 semi-structured interviews were completed at four weeks and four months after hip fracture. Interviews were conducted with the patients themselves, their carers or in some cases with patient/carers both present. Analysis found first, that restoration of mobility was important, and second, that the hip fracture was viewed as part of the general decline in health with age. The authors interpreted the data to conclude that patient reported outcome measures of general health-related quality of life may provide useful information for most of these patients. The methods section could benefit from including detail about the interview schedule, methods for enhancing rigour, and a more explicit description of the qualitative approach to analysis. The second part of the analysis was not well explained, seemed at odds with the thematic analysis completed in the first part, and did not appear to directly address the second research question – it detracted from rather than enhanced the manuscript. Finally, the interpretation that generic quality of life scales should be used to address patients perceived limitations in mobility does not follow. Some specific comments: Abstract (page 3, line 20): The second approach to analysis: 'extraction of data into template' is not clear. Abstract (page 3, line 33): What is 'stable mobility'? Abstract (page 3, line 51): Key messages (page 4, second dot point): How does a general health-related quality of life tool address the perceived issue of limited mobility in this group? Introduction (page 6, line 10): Would the qualitative research design be expected to provide data to address the second research question on variation? Method (page 7, about line 57): Since the interviews were semi-structured, please include the interview schedule with sample questions and prompts. Method (page 8, about line 24): For qualitative research it is important to be explicit about your approach to data analysis. While understanding that this can be difficult with terminology and definitions (e.g. grounded theory means different things to different researchers), it is important to describe whether you applied existing theory when you coded or whether you explicitly coded what emerged from the data, trying not to impose existing ideas or theories. This has to be addressed. Method (page 8, line 33): It is a concern that only one in four transcripts was coded by two researchers. Method (page 8, about line 37): The analysis approach to address the second research question is not clear. It appears that the template was developed prior to looking at the interview transcripts and then the transcripts were assessed to see how they fitted with the pre-determined categories. There are a number of concerns
--

	here: first, it is not clear how this approach addresses the stated aim of exploring variation; second, the approach of this second stage where what participants said is made to fit with the template appears at odds with the first stage where the themes emerged. I question whether this second stage detracts rather than adds to the analysis. Methods (page 8, about line 50): What methods were used to enhance trustworthiness/rigour of the data and analysis? Results (page 9, line 5): Carers will have a different perspective than the people who have experienced the hip fracture. This needs to be acknowledged in the Discussion. Results (page 13, about line 43): Did the theme of perceiving the fracture as part of the process of ageing emerge for stage 1 of the analysis or from stage 2? Discussion (page 16, paragraph 2): The rationale for recommending generic quality of life scales as being appropriate to address the perceived limitations in mobility does not make sense to me.
--	--

VERSION 1 – AUTHOR RESPONSE

We appreciate the time and care taken by the reviewers of our manuscript. Below we set out our responses to the reviewer’s comments. We provide the text of each reviewer comment followed by our response (to make it clear which text is our response, we start each response with ‘We’).

Please include the study design in your title.

We have specified in the title that this is an interview study.

This study addresses an important issue related to wellbeing and evaluation of services to hip fracture patients. The objectives of this study are to (1) explore what is important to patients in postacute functional recovery among hip fracture patients who received surgical repairs, and (2) consider how these reported priorities could be used in advancing the evaluation of the quality of hip fracture services. The manuscript is confirmatory and may add new information to the hip fracture functional recovery literature from the perspectives of the patients, carers, and patients with dementia. Major revisions, however, may be needed to address the research questions.

Minor issues

1. Literature updates: most of literature cited in this study were in or prior to 2012; only one article was in 2013. More current literature should be included.

We have searched the literature for more recent studies. There is literature published in 2013 onwards on hip fracture and patient experience. This includes the reviewer’s own paper on multiple morbidity and outcomes (1). However, we did not find recent literature that related specifically to our study topic for adding to our discussion section.

2. Figure 1 clarification: the flowchart does not provide a good summary of the Participant section. Figure 1 starts with the number of recruitments of study sample and ends with number of interviews. The following clarifications may be helpful for readers to follow the sample selection: a) The final number of study subjects was 31. Of these, how many interviews were conducted with individual patients, with individual carers, and the number of joint interviews conducted with carers responding for patients with dementia or too frail to be interviewed. b) Separation of those who were interviewed at 4 weeks or 4 months only, and those who were interviewed twice at 4 weeks and 4 months (n=10).

We have considered the above comment carefully. Figure 1 is designed to clarify for the reader the recruitment process for this study. The details mentioned by the reviewer above are about post

recruitment. We feel this is better described in the text, as it currently is.

Major issues

The study combined the data collected at 4 weeks and 4 months because the self-reported results are similar (page 8, Analysis, lines 15–20). The results from this combination of time related to functional recovery at approximately 4 weeks and 4 months may lead to biased results and may be less informative to address the second objective of the study. The data analysis plan may need to be changed to address the following issues: 1. Separate analysis of functional recovery at 4 weeks vs. 4 months may be needed. Literature have shown that hip fractures have significant impacts on functioning related to the ability of self-care (ADLs) and independent living in the community (IADLS). These adverse outcomes mainly relate to compromised mobility function as a result of hip fractures. The process of regaining prefracture function can be lengthy and stressful. Most hip fracture patients were still very disabled at 4 weeks following hip fracture surgery and intensive postacute rehabilitation. Substantial peak function recovery was observed at approximately 4 months following postacute rehabilitation (a lot of literature available in Medline/Pubmed search), thereafter, the functional recovery slows down and level off. Lumping together data collected at 4 weeks and 4 months is less informative to study objective 2; that is, the priority considered important to the patients reported by the study can be used in the evaluation of the quality of hip fracture services (page 3, lines 10-11).

We have considered the above comments carefully. We are aware of the difference in functioning at four weeks vs four months post hip fracture and we comment that this was apparent in the interview data. However, the themes identified by the participants were similar at four weeks and four months. We have clarified this in the first paragraph of the analysis section.

2. Separate data analysis by patients and carers.

Studies on validity and reliability of self-reported data have shown that bias may result from self-reported data. For example, hip fracture patients often underestimate their functional disabilities; on the other hand, carers—often very protective—overestimate the level of disability of the patient.

We agree that this is important when assessing outcome. However, our aim was to understand what was important to participants during recovery rather than the amount of recovery. We found carers and participants talked about similar themes. We have clarified this in the discussion.

3. A homogeneity study may be performed to enhance generalizability of the study. The authors mentioned in the study Limitations section that patients who were unable to participate in the study may be older, sicker, and more disabled. Homogeneity tests on age, gender, selected health, or functional status may be performed to address whether the participants and non-participants were indeed different.

We consider this question in the manuscript by comparing age, gender and cognitive impairment for the interview sample with UK data (see first paragraph of section on strengths and limitations). This provides evidence as to the generalisability to the hip fracture population in the UK. We think readers will want to know this, rather than a comparison of interview study participants with the hip fracture population who enter the cohort study within which this study was nested.

4. Results and interpretation should focus on addressing the two study objectives: a) to clearly present what you found from the patients, carers, or patients/carers related to the time frame in which data are collected – what results are confirmatory and what are contributions of this study to the literature.

We have a section in the discussion section on comparison with other studies. In this we explain the

consistency of our results with other studies. We use the word 'consistent' as in qualitative research there are always differences in detail and emphasis in results even where the findings are very similar.

b) the second objective of the study was less articulated.

We have edited this to clarify.

In page 16 lines 22-24 "we now have sufficient qualitative evidence to inform policy decisions about the choice of appropriate PROMS for assessing recovery from hip fracture." The authors may want to quote specific evidence to support the statement.

We have changed 'we now' to 'there is' to clarify. The research discussed in this section has produced similar results to ours so taken together they provide the evidence for this statement.

Other statements in Lines 33-35 and lines 54-56 in page 16 are appropriate for the study results: "it is unlikely that a single PROM specific to hip fracture could be developed which is relevant to the whole spectrum of patients," "...it may be impossible to devise a single PROM that will be appropriate for all patients," respectively. Many of PROMS measures were originally designed to assess intervention or treatment effectiveness in clinical trials, but are now used more extensively to assess patient perspectives of care outcomes. This study results provide useful information to suggest that once a seemingly appropriate measurement has been identified, it is advisable to pilot the questionnaire on a small number of patients given the heterogeneity of hip fracture patients and the diverse priority list on recovery concerns generated from this study.

We thank the reviewer for the above comments which are supportive of our conclusions.

Evaluating recovery following hip fracture: a qualitative study of what is important to patients This qualitative study explored what patients and their carers consider important when evaluating their recovery from hip fracture. A total of 41 semi-structured interviews were completed at four weeks and four months after hip fracture. Interviews were conducted with the patients themselves, their carers or in some cases with patient/carers both present. Analysis found first, that restoration of mobility was important, and second, that the hip fracture was viewed as part of the general decline in health with age. The authors interpreted the data to conclude that patient reported outcome measures of general health-related quality of life may provide useful information for most of these patients. The methods section could benefit from including detail about the interview schedule, methods for enhancing rigour, and a more explicit description of the qualitative approach to analysis. The second part of the analysis was not well explained, seemed at odds with the thematic analysis completed in the first part, and did not appear to directly address the second research question – it detracted from rather than enhanced the manuscript. Finally, the interpretation that generic quality of life scales should be used to address patients perceived limitations in mobility does not follow.

Some specific comments:

Abstract (page 3, line 20): The second approach to analysis: 'extraction of data into template' is not clear.

We have edited to clarify.

Abstract (page 3, line 33): What is 'stable mobility'?

We have edited to clarify.

Abstract (page 3, line 51): Key messages (page 4, second dot point): How does a general health-related quality of life tool address the perceived issue of limited mobility in this group?

We have considered the text again and feel it sufficiently explains that mobility is important for many other domains in quality of life tools.

Introduction (page 6, line 10): Would the qualitative research design be expected to provide data to address the second research question on variation?

We have edited the second research question to clarify. Qualitative data can provide evidence about variation. We have added a sentence at the end of the methods section to clarify this.

Method (page 7, about line 57): Since the interviews were semi-structured, please include the interview schedule with sample questions and prompts.

We have six variations on the interview schedule so rather than providing the actual interview schedules as supplementary material, we have inserted into the text the key questions asked in each interview. This has enabled us to clarify for the reader which questions were used for the different sections of the interview.

Method (page 8, about line 24): For qualitative research it is important to be explicit about your approach to data analysis. While understanding that this can be difficult with terminology and definitions (e.g. grounded theory means different things to different researchers), it is important to describe whether you applied existing theory when you coded or whether you explicitly coded what emerged from the data, trying not to impose existing ideas or theories. This has to be addressed.

We have clarified in the analysis section, first paragraph, that we identified the issues specific to hip fracture recovery from the data itself. We have further clarified this in the second paragraph by explaining that we examined and coded what the participants offered as important to recovery.

Method (page 8, line 33): It is a concern that only one in four transcripts was coded by two researchers.

We have added more detail so the reader is clear about the details of the analysis process. All transcripts were read by at least two members of the research team and their notes from this reading brought to our analysis meetings. In this way all data was examined by two members of the research team. Formal coding in NVivo for the first analysis approach was undertaken on one in four transcripts to check consistency of this coding. We have clarified this in the analysis section paragraphs one and two.

Method (page 8, about line 37): The analysis approach to address the second research question is not clear. It appears that the template was developed prior to looking at the interview transcripts and then the transcripts were assessed to see how they fitted with the pre-determined categories. There are a number of concerns here: first, it is not clear how this approach addresses the stated aim of exploring variation; second, the approach of this second stage where what participants said is made to fit with the template appears at odds with the first stage where the themes emerged. I question whether this second stage detracts rather than adds to the analysis.

We are grateful for the reviewer sharing this concern as it alerted us to an error we made in the manuscript. In paragraph three of the analysis section we should have written that the analysis summarised the experience of hip fracture recovery for each patient carer dyad. Similarly this should be reflected in the second research question and the relevant results section. We have corrected this

error.

We have clarified that the second research question and analysis approach arose from the first data analysis. We have clarified that the template was developed from the data itself and was not pre-specified. (paragraphs 2 and 3 of analysis section)

Methods (page 8, about line 50): What methods were used to enhance trustworthiness/rigour of the data and analysis?

We have clarified in paragraph one of analysis section, that transcripts were read by at least two team members. We have included more detail about the role of team analysis meetings throughout the analysis section. The text already included details of more formal analysis quality checks.

Results (page 9, line 5): Carers will have a different perspective than the people who have experienced the hip fracture. This needs to be acknowledged in the Discussion.

We have clarified in the discussion section that carers talked about the same themes as those experiencing hip fracture.

Results (page 13, about line 43): Did the theme of perceiving the fracture as part of the process of ageing emerge for stage 1 of the analysis or from stage 2?

We have clarified this in the section now entitled: 'Is there variation in the experience of what is important in recovery from a hip fracture?'.

Discussion (page 16, paragraph 2): The rationale for recommending generic quality of life scales as being appropriate to address the perceived limitations in mobility does not make sense to me.

We have edited the conclusion of the abstract to more accurately reflect the conclusion of the paper. We have removed the words 'our more active' in this paragraph as it may be misleading. The paragraph about use of PROMS concludes by suggesting these might be appropriate for clinical trials, i.e. where there is a control group. The subsequent paragraph discusses the use of PROMS for assessing quality of care – the focus of this paper. Here we state that 'it may be impossible to devise a single PROM that will be appropriate for all patients'.

1. Mathew R, Hsu W-H, Young Y. Effect of Comorbidity on Functional Recovery After Hip Fracture in the Elderly. American Journal of Physical Medicine and Rehabilitation. 2013;92(8):686-96.

VERSION 2 – REVIEW

REVIEWER	Nicholas Taylor La Trobe University, Australia
REVIEW RETURNED	16-Jul-2014

GENERAL COMMENTS	The authors have addressed some issues in the revised manuscript but there remain some significant concerns: 1. The second research question, the approach to analysis in part 2, and results in part 2 remain unclear and do not add to the interpretation of the data. a. First, the main themes that emerged from the data appeared to emerge, or should have emerged from the coding and thematic analysis conducted in part 1 of the study. That is, that mobility is important, and that the hip fracture was viewed as part of a general decline in health with age. It is not clear what 'variation' was in fact
--

	captured by this second analysis. b. Second, the approach to analysis for the second part of the study appears very unusual, in a way circular, and not designed to address the second research question. The authors describe that based on the first five transcripts, a draft template was developed to summarise living arrangements and the environment, day to day life now and in the past, the impact of the fractures and its management, what was changing as they recovered, the extent to which the patient referred specifically to the fracture, and their ability to engage in the interview. The data from each patient was then summarised, and the researchers then compared the summaries. Fundamentally this approach in part 2 is different from part 1, where the themes emerged, since patient interview data was made to fit into the structured summaries (after the first five interviews). Also, it appears as though the analysis was completed on the summaries rather than the interview data from the patient/carer – which could be a problem as it involved the researchers introducing their own biases. Having a clear theoretical framework for how the qualitative analysis would be conducted may have avoided this problem. 2. It remains important for the researchers to report specifically on strategies they used to enhance the rigour and trustworthiness of their data and it is not sufficient to add a couple of details e.g. that at least two researchers read each transcript. How were credibility, transferability, dependability, and confirmability addressed (for example, see the McMaster qualitative critical appraisal tool)? 3. If the main theme that emerged was that mobility is important, then surely an outcome measure that captures the construct of mobility in patients with hip fracture is more relevant than a general health-related quality of life (HRQOL) measure. Therefore, the conclusions that the results suggest routine use of HRQOL measure in this population does not seem to be supported by the data. If mobility is the most important construct for patients recovering from hip fracture then it would follow that a measure that captures that construct (and is not diluted by other domains within HRQOL measures) is important. Also, if qualitative analysis has identified what is of concern to patients recovering from hip fracture, then why does it follow that the outcome measure has to be a PROM?
--	--

VERSION 2 – AUTHOR RESPONSE

Reviewer comment

The authors have addressed some issues in the revised manuscript but there remain some significant concerns:

1. The second research question, the approach to analysis in part 2, and results in part 2 remain unclear and do not add to the interpretation of the data.

Author response

We have added text to the introduction to clarify the reason for the second research question. We have edited research question 2 to clarify that we were not only interested in variation within the population but the reasons for this variation. In the analysis section of the method we have added text and references to clarify that the second analysis approach is cross-case comparison. This approach is used to detect variation between individual cases and understand any variation found.

Reviewer comment

a. First, the main themes that emerged from the data appeared to emerge, or should have emerged

from the coding and thematic analysis conducted in part 1 of the study. That is, that mobility is important, and that the hip fracture was viewed as part of a general decline in health with age. It is not clear what 'variation' was in fact captured by this second analysis.

Author response

We have clarified in the analysis section of the methods that the first approach to analysis identified themes for the whole group of participants. This is in the very nature of thematic analysis and we provide a reference. We agree with the reviewer that mobility was important to all participants. However, not all participants considered hip fracture as part of a general decline in health with age.

We point you to the follow section of text in the results:

those who were active pre-fracture talked in terms of regaining a recovered state that was similar to their pre-fracture state although with some minor adaptations (participants 15 and 20 in box 1). Whilst these participants expressed worry about how well they might function in the future, there was, nevertheless, determination to progress to as full a recovery as possible. Four months post-operation many of these participants had all but regained their pre-fracture level of activity.

We also point to the summary of participant 15 and 20 in the table.

Reviewer comment

b. Second, the approach to analysis for the second part of the study appears very unusual, in a way circular, and not designed to address the second research question. The authors describe that based on the first five transcripts, a draft template was developed to summarise living arrangements and the environment, day to day life now and in the past, the impact of the fractures and its management, what was changing as they recovered, the extent to which the patient referred specifically to the fracture, and their ability to engage in the interview. The data from each patient was then summarised, and the researchers then compared the summaries. Fundamentally this approach in part 2 is different from part 1, where the themes emerged, since patient interview data was made to fit into the structured summaries (after the first five interviews). Also, it appears as though the analysis was completed on the summaries rather than the interview data from the patient/carer – which could be a problem as it involved the researchers introducing their own biases. Having a clear theoretical framework for how the qualitative analysis would be conducted may have avoided this problem.

Author response

In the analysis section of the methods, we have clarified in the text that the second approach to analysis is what can be termed cross-case analysis. We have provided references to standard texts on this approach. As this approach is most commonly used for studying cases such as organisations or teams, we have added text to clarify that the 'case' in this study is the individual recovering from hip fracture and referenced literature where theory of the 'case' is discussed. We have also clarified that the template summaries for each case form a matrix for cross-case analysis. We have clarified that the template was refined based on the data so the reader is clear the data was not made to fit the template. For cross case analysis it is normal practice to use summaries of raw data in the matrix used for comparison.

We appreciate the reviewer raising the issue of theory. In the analysis section of the methods, we have reinserted text that we had written in an earlier, pre-submission, version of the paper, on our theoretical approach to data collection and analysis. This is in addition to the clarification of the nature of the 'case' with references described above.

Reviewer comment

2. It remains important for the researchers to report specifically on strategies they used to enhance

the rigour and trustworthiness of their data and it is not sufficient to add a couple of details e.g. that at least two researchers read each transcript. How were credibility, transferability, dependability, and confirmability addressed (for example, see the McMaster qualitative critical appraisal tool)?

Author response

We have checked our paper carefully against the McMaster qualitative critical appraisal tool. By undertaking this process we realised that there were aspects of our research process that could be made clearer for the reader. We have described the interdisciplinary nature of the team; we have clarified the nature of the refinements to the interview schedule, we have clarified that analysis commenced during recruitment so we could judge when we had reached data saturation; we have clarified that the researchers made reflective field notes.

Reviewer comment

3. If the main theme that emerged was that mobility is important, then surely an outcome measure that captures the construct of mobility in patients with hip fracture is more relevant than a general health-related quality of life (HRQOL) measure. Therefore, the conclusions that the results suggest routine use of HRQOL measure in this population does not seem to be supported s by the data. If mobility is the most important construct for patients recovering from hip fracture then it would follow that a measure that captures that construct (and is not diluted by other domains within HRQOL measures) is important. Also, if qualitative analysis has identified what is of concern to patients recovering from hip fracture, then why does it follow that the outcome measure has to be a PROM?

Authors response

We have edited the discussion section of the paper to address these issues. We have also edited our abstract and key points.

VERSION 3 - REVIEW

REVIEWER	Nicholas Taylor La Trobe University, Australia
REVIEW RETURNED	30-Oct-2014

GENERAL COMMENTS	The authors have addressed the review comments resulting in an improved manuscript. The description of the analysis for the second part of the study is clearer and appropriately cited. The discussion about an appropriate PROM to use in this population is now more balanced. Finally, some details have been added throughout the manuscript, which would aid the reader in evaluating the rigour and trustworthiness of the data.
---